# Language of Health of Young Australian Adults: A Qualitative Exploration of Perceptions of Health, Wellbeing and Health Promotion via Online Conversations

**DOI:** 10.3390/nu12040887

**Published:** 2020-03-25

**Authors:** Annika Molenaar, Tammie S. T. Choi, Linda Brennan, Mike Reid, Megan S. C. Lim, Helen Truby, Tracy A. McCaffrey

**Affiliations:** 1Department of Nutrition, Dietetics and Food, Monash University, Notting Hill 3168, Australia; annika.molenaar@monash.edu (A.M.); tammie.choi@monash.edu (T.S.T.C.); helen.truby@monash.edu (H.T.); 2School of Media and Communications, RMIT University, Melbourne 3000, Australia; linda.brennan@rmit.edu.au; 3School of Economics, Finance and Marketing, RMIT University, Melbourne 3000, Australia; mike.reid@rmit.edu.au; 4Behaviours and Health Risks, Burnet Institute, Melbourne 3004, Australia; megan.lim@burnet.edu.au; 5Melbourne School of Population and Global Health, University of Melbourne, Carlton 3053, Australia

**Keywords:** young adults, university students, eating behaviours, physical activity, mental health, wellbeing, health communication, health promotion, qualitative methods

## Abstract

Young adults (YA) are in a critical stage of life for the encouragement of healthy behaviours such as healthy eating and exercising. This research explored Australian YA values and perceptions related to health, healthy behaviours and health promotion efforts. This qualitative analysis involved *n* = 166, Australian 18–24 year-olds recruited through a market research field house. Participants (63% currently studying at tertiary level) engaged in a facilitated in-depth online conversation around health and healthy eating over four weeks. Leximancer^TM^ and manual inductive thematic coding were utilised for analysis via the lens of emerging adulthood and health communication theories. Health was seen as multi-faceted, with particular importance placed on mental health and exercise. Some participants focussed on physical appearance, often fuelled by comparison to others. Many believed that at their age and health status, adopting health-enhancing behaviours without short-term tangible benefits was not a priority. Participants did not prioritise health-enhancing behaviours due to barriers such as a perceived lack of money, knowledge and time often due to studying or working and perceived effort. Strategies they proposed to encourage healthy eating included: incentivising healthy food; quick and affordable healthy recipes; and communicating the short-term tangible benefits of healthy behaviours. There is a need for focussed health messaging that address the needs and desires of YA and directly address the barriers they face.

## 1. Introduction

Young adulthood is a distinct and pivotal stage in one’s life where lifelong values and habits such as those related to health-enhancing behaviours are developed and sometimes cemented [1]. However, young adults (YA) including university students, are often an overlooked age group in relation to their health, wellbeing and health care. Compared to other age groups, YA have amongst the lowest utilisation rates of health care [2] and may not be receiving the preventative health care they need to encourage health-enhancing behaviours to optimise their health now and into the future [3]. YAs have been found to be in worse health than adolescents [4] and people in their late 20s and 30s [5]. In this age group, mental health illnesses are amongst the leading causes of burden of disease [6]. During the transition from adolescence to adulthood, overall diet quality appears to reduce [1,7], and compared to both younger and older age groups, YA consume less fruit and vegetables [8] and more energy-dense-nutrient-poor fast food [9,10]. Physical activity has also been found to decline during the shift into adulthood from adolescence [1,11] and appears to continue to decline with age [12]. These factors contribute to the increasing prevalence of weight gain in YA [13] particularly in the first years of tertiary education [14] and the subsequent risk of developing chronic diseases later in life [15]. Therefore, it is important to guide YA to adopt healthy lifelong habits to ensure optimal health status in later life.

The process of developing independence and beginning ‘adulthood’ has become prolonged due to cultural, social and economic forces such as increased numbers of individuals seeking tertiary education, increased costs of education and living, a lack of job opportunities, and greater control over the timing of life events, e.g., marriage [16]. This has been described by Arnett et al. as Emerging Adulthood theory [17] and is characterised by the development of identity and values related to different areas of life including love, work, worldviews and risky behaviours. We postulate that this could include the value placed on health-enhancing behaviours and the formation of habits related to these behaviours such as substance use, eating and physical activity; therefore, targeting this potentially malleable age group could be an impactful health promotion effort [1].

Health communication theory posits that the promotion of health information to educate and empower individuals to improve their health, requires communication that is easily understood, on an accessible and appropriate platform and that resonates with the target group [18]. One such platform is social media (SM), which, as it is viewed as an acceptable method of information delivery [19,20], could be an effective way forward for health communication. Relative to commercial brands, organisations involved in health promotion have generally been slower to adapt to the ever-evolving SM environment. SM has the potential to support health behaviour change by enhancing knowledge and self-efficacy through SM’s extensive reach, consumer engagement and accessibility [21]. As health communication theory suggests, in order to successfully influence an audience, there is a need to understand the needs and desires of the target audience and the barriers they may face to adopting the intended behaviour [18].

Research on overall health and wellbeing in YA and university students is limited with a focus more on specific aspects of health, e.g., alcohol use, drug use and sexual health [22] or healthy eating [23]. Therefore, the primary objective of this research was to explore what health, wellbeing and healthy eating mean to a sample of Australian YA aged 18 to 24 years old who were primarily completing tertiary education recruited through a market research field house, and what aspects of health are important to them at this stage of their lives. A secondary objective was to explore these YA perspectives of a current health communication strategy and allow them to develop ideas of potential future health promotion campaigns to build an understanding of how YA believe health promotion campaigns could resonate and meet their needs and desires. The purpose of this research was to provide formative information to inform future more hypothesis-driven research and was therefore exploratory in nature.

## 2. Materials and Methods

### 2.1. Online Conversations

These analyses report qualitative findings from two forums and one challenge where participants are asked to think of a potential solution to a posed issue, out of a total of 20 forums, two challenges, three polls and one journal entry from within the formative Phase 1a of the ‘Communicating Health Study’. Details of the Communicating Health study as a whole can be found here [24]. In brief, Communicating Health is a four-year study applying social marketing techniques in order to understand how YA engage with health and healthy eating, particularly on SM. The outcomes of the overall study will inform the creation of engaging social marketing messages that motivate and engage YA. Phase 1a involved a four-week online conversation with YA discussing a series of health, SM and eating-related topics, operated and moderated by an independent market research field house. Online conversations utilise the principles of digital ethnography [25] and involve the use of an online community methodology to obtain rich insights into consumer behaviour [26]. Online conversations involve participants in the communities’ conversing through forums over a longer period than is taken by other qualitative methods, such as interviews and focus groups. The following sections describe the overall procedure for Phase 1a with specific details related to the forums and challenge in these analyses.

Phase 1a received ethics approval from the RMIT Business College Human Ethics Advisory Network (Project number: 20489) and Monash University Human Research Ethics Committee (Project number: 7807). Participants consented to anonymised findings being published when they completed the patient information and consent form prior to participating in the study.

### 2.2. Recruitment

For Phase 1a the recruitment target was set at *n* = 200 YA (aged 18–24 years) to achieve an extensive data source based on previous work using a similar methodology [27]. Participants were recruited by an Australian Market & Social Research Society-certified field house [28] from three different International Organization for Standardization accredited panels [29] to ensure a wide mix of people across the sample as well as to reach quotas. The sampling quotas for this study were set to be approximately representative of the Australian population [30] on gender and location (both Australian State or Territory and location type, i.e., metropolitan and regional locations).

Panel members were from a mix of different panels, were already part of the field house’s database, and voluntarily provided their details in the expectation they would be invited to partake in research. The field house recruited panel members for the purpose of research, not specifically for this study. Research panels are recruited through online advertising and social media targeted at a range of individuals—not specifically YAs—and anyone is eligible to sign up to a panel. The online conversations study was advertised to those within these groups through online panel research websites. They were emailed and invited to complete a screening questionnaire to assess their eligibility [31]. People were eligible if they were aged between 18–24 years, currently residing in Australia and self-reported using SM at least twice a day. Eligible people were emailed a profiling and registration survey within a week, including demographic information, self-reported weight, height, SM use and interest in health [31].

Completers of this profiling survey who registered by creating log in details on the online conversations’ website were assigned to one of four online communities based on their age and interest in health. The profiling aimed to achieve four approximately equal online communities based on age; 18–21 or 22–24, and interest in health; low or mid/high. Interest in health classification was based upon the median value for the following question from the profiling survey: “On a scale of 1–7, where 1 means “Strongly Disagree” and 7 means “Strongly Agree”, please indicate how strongly you agree with the following statement—“I take an active interest in my health.”

The dropout rate (Figure 1) was as expected for this age group [32,33]. This led to the implementation of a referral system, whereby participants could invite their friends (who were then screened and profiled in the same way). The incentive for participating was an AUD 100 gift voucher, and the 20 most comprehensive contributors (i.e., five participants per online community) received an additional AUD 100 voucher.

### 2.3. Data Collection

The online conversations were presented through an online ‘lounge’ hosted on a private website over four weeks (10th May to 6th of June 2017). The online conversations included 20 different forums and two challenges which were designed to take up to five minutes each to complete (total of approximately 110 min), three short polls and an ongoing journal entry which participants had to contribute to at least four times [31]. Participants were expected to complete everything to receive the incentive. The forums were released at different times throughout the four-week period, however remained open throughout to contribute to and interact with others.

Two moderators facilitated the online conversations and asked follow-up questions. Moderators included both a male (Masters of Management Marketing/Finance) and female (Bachelor of Psychology & Sociology, Masters of Applied Social Research) moderator, both with experience in market research. As the moderators were unknown to the participants, an introductory forum was used to build rapport. Data were collected in the form of text responses and uploaded images in response to questions.

As outlined, these analyses present two of the forums and one challenge. These were chosen based on their discussion of health and health messaging. Discussion guides and the rationale for these specific forums can be found in Table 1. Due to the forums being released over different weeks, there are different completion rates (Figure 1).

### 2.4. Data Analysis

Different qualitative data analysis approaches were adopted for the two forums and for one challenge intended to draw out meaningful messages. Leximancer^TM^ 4.5 [34], a computer-assisted content analysis program, was used to provide a ‘big picture’ concept of the data set, highlighting key concepts created based on high-frequency keywords for focussed manual thematic analysis. Leximancer^TM^ applies an algorithm to identify concepts based on word frequency statistics and co-occurrences and generates groupings of similar text excerpts without any pre-existing assumptions of the meaning or context of the text [35,36]. Leximancer^TM^ produces concept maps which visually cluster frequent keywords and categorises into concepts which are displayed in relation to each other and with a ranking of importance [37]. As Leximancer^TM^ does not consider the meaning behind the key concepts, the first 100 text extracts for the top 10 high frequency concepts from both Forum 4 and Forum 14 were manually thematically analysed by AM and TC (total of 2000 text extracts). Challenge 1 was analysed solely using manual inductive thematic analysis [38] to gain insight into the individual health campaigns and messages proposed in the context of the participant’s whole response.

While the forums and challenges were discreet, they were interpreted collectively in the development of overall themes to answer both objectives. The thematic analysis utilised investigator knowledge to interpret the data via the lenses of two existing theories: emerging adulthood theory [17] and health communication theory [18]. Investigator triangulation was employed [39] as the analysis of all forum responses was conducted by two female authors (AM and TC) independently before coming together to discuss and coming to a consensus. The analysis was conducted by AM—a nutritionist (Bachelor of Nutrition Science Honours), and TC—a dietitian (PhD), who both have experience in qualitative research. Their knowledge of nutrition, the Australian Guide to Healthy Eating (AGHE) and nutrition campaigns influenced their interpretation and the meaning derived from the forum conversations. As AM is similar in age to the participants, there was ability to interpret findings from the lens of an emerging adult.

## 3. Results

### 3.1. Participant Characteristics

A total of 163 YA completed at least one of the forums/challenges included in this analysis. Participants were primarily Female (60.7%) and healthy weight (54.0%) (Table 2). Overall participants were generally well educated with the majority (62.6%) currently studying a tertiary degree. Disposable income was low for this group with 39.3% having less than AUD 40 (EUR 27, USD 35) a week. Most participants were currently living with their parents or family.

### 3.2. Thematic Analysis

The Leximancer^TM^-generated concept maps for both Forum 4 and Forum 14 (Appendix A
Figure A1 and Appendix A
Figure A2) graphically demonstrate high-frequency key concepts within those forums, which were used to focus the manual thematic analysis. Themes included: (1) perceptions of health and wellbeing through the lens of emerging adults; (2) competing demands and priorities limiting health-enhancing behaviours; (3) transformation of identity in emerging adulthood; (4) promotion of health-enhancing behaviours needs to be different and tailored (Table 3).

#### 3.2.1. Theme 1: Perceptions of Health and Wellbeing through the Lens of Emerging Adults

Participants often described health and wellbeing as a balance between their physical, emotional, financial, social, cultural and spiritual states. The majority of participants in these analyses believed they were at their optimal health status, and were therefore not largely concerned about illness or disease prevention but rather a more holistic view of health.

“To me health or well being primarily means physical health, good eating and adequate exercise, but also is inclusive of social health (e.g., friendship connections, relationships with relatives, frequent contact with people and the community) and mental health, such as having time to relax, sleep, maybe mediation, as well as state of mind...” (Forum 4: Female, 18 years old)

In particular, mental health and having an optimal mental status was highlighted for its importance in contributing to health and wellbeing, often due to the pressure they felt to balance so many aspects of their lives such as university and work. Many YA were personally affected or knew peers who were affected by mental illness, which resulted in a reduced stigmatisation of mental illnesses.

“Myself and a few of my friends suffer from mental health issues and that has always challenged our ability to achieve our desired physical and mental health. I think about my mental health more than my physical health, because how I feel always overcomes the way in which I take care of my body.” *(Forum 4: Female, 20 years old)*

Having friends that one could relate to and that were there as a support system was an important contributor to social, mental and overall health and wellbeing. Being able to spend time with friends and having meaningful connections with others were a priority in many YA lives and contributed to their wellbeing.

Participants also described healthy eating and exercise as enhancing their health and wellbeing; however, individuals were at different stages of engaging in these behaviours. Some described an awareness, responsibility for and active incorporation of healthy behaviours into their lives, and others aspired to be healthier but felt limited by the environment they were in or presented with low motivation and perceived desire to change.

“I think about my health constantly. Yet despite this I continue to do nothing about it. I’m always hoping that once my study workload makes way for a better social life that I will commit to being more fit but I find it hard to put it as a priority, which is frustrating because it should be!” (Forum 4: Female, 18 years old)

It was difficult for these YA to relate to lifestyle illnesses they might develop in their later lives. They reported a preference and understanding of short-term, often gender-specific, tangible consequences of health-enhancing behaviours particularly exercise or physical activity. Males were often concerned with being physically strong, while for females it was more about being slim and toned. Being physically active was also thought to make you feel better both physically, such as increased energy, and mentally.

“Now that I am training [at the gym], it’s been great! I still have a lot to work on but I am more focused and healthy and energised! I feel I have a lot of strength to do certain things! It is a great feeling!” (Forum 4: Male, 20 years old)

Having described the patterns of perceptions of health among the YA participants, the diversity needs be acknowledged. It was apparent that some younger participants in the study (i.e., 18–21 year-olds) focussed more on the physical weight and body appearance aspects of health. Being healthy, to them, involved looking socially attractive, fit and lean for women or muscular for men. This was not exclusive to this age group and was not encompassing of all participants, but there appeared to be a trend towards this way of thinking in this age group. On the other hand, some older YA (i.e., 22–24 year-olds) described a change in their view of health as they grew older.

“I feel like ‘health and well-being’ is different for young adults compared to older people, as I feel like young adults tend to focus more on physical health than the other health aspects. Especially with dealing with pressure to do well at ‘uni/work’ and ‘being attractive’, young adults spend more time eg. at the gym and food prepping, rather than taking a moment to relax and stress their mind with everything going on, or distancing themselves from everyone else just to achieve their goals.” (Forum 4: Female, 19 years old)

#### 3.2.2. Theme 2: Competing Demands and Priorities Limiting Health-Enhancing Behaviours

In general, participants reported feeling powerless when it came to adopting certain healthy behaviours into their lives. Being healthy, eating healthily and exercising were often aspirations, and they felt they could not consistently perform these behaviours in their current lives due to various social, environmental and personal barriers. One major barrier was the cost of eating healthily, exercise and health care. “Healthy” foods—especially perishable fresh fruit and vegetables—were seen as particularly expensive. Participants reported energy-dense, nutrient-poor convenience foods were cheaper than purchasing and preparing a meal with fresh ingredients. Exercise, including gym memberships and team sports, was also deemed to be expensive by some individuals. Some mentioned they would like to exercise and participate in sport, but due to low income and recent financial independence, they currently could not afford to.

“Where as when you were a child your mother or father pick what you eat and do. How it changed over the years.... Well I had to start by paying for it myself so I had to be a bit more selective on what I did and how much it cost so I didn’t get to do or eat everything I did as a kid because I can’t afford it.” (Forum 4: Female, 23 years old)

These YA repeatedly described personal and social barriers—such as being time-poor—with study, work and socialising taking up the majority of their time. Many mentioned working on top of studying or working multiple jobs, which limited their time for health-enhancing behaviours such as cooking and exercising.

“For me, it means balancing work, uni, socialising with friends and family and still feeling okay. I think for young adults, we tend to be very aware of it [Health], but not necessarily doing anything about it.” (Forum 4: Female, 19 years old)

Time constraints—along with the perceived rise in food delivery and ease of access in settings such as university—were a common reason as to why they chose to eat convenience foods and takeaway.

“Now busy at uni and work I maybe run around once a week and buy more fast food meals than I’d like to think about, and I have noticed a physical toll from this on my body.” (Forum 4: Female, 20 years old)

Socialising also involved the consumption of fast food or eating out with friends and drinking alcohol, which reduced their ability to prepare healthy food. There was also a sense of a lack of willpower to say no to certain “unhealthy” convenience foods. Overall, although they knew they “should” prepare healthy food, many lacked the motivation to do so or viewed it as a burden.

“I always cook healthy food but where I go wrong is being tempted by the convenience of not cooking, because those foods are always unhealthy takeaway. That one is difficult in the sense of finding the motivation, but easy in the sense of knowing what to do.” (Forum 4: Female, 24 years old)

The media, including communications from the Government, and SM influencers, were seen to perpetuate misconceptions and conflicting messages of what is healthy which sometimes reduced YAs understanding of what a healthy diet looks like. Government messages such as the AGHE were often viewed as wrong or hard to believe, as it did not correlate with what they “knew” or other messages online. Some participants who followed an alternative diet to the AGHE, particularly those following a fad diet, did not trust the AGHE messages. These individuals had strong beliefs that their way of eating was what is “actually healthy”, specifically the exclusion of certain food groups—commonly carbohydrates and dairy. People sometimes viewed the guide as “one-size fits all” and not compatible with everyone and their different and specific needs—particularly those on a specific diet, e.g., plant-based, or those with certain allergies.

“I’d say it’s (the AGHE) an unrealistic representation of the “perfect diet” according to dietitians and doctors. Everybody’s diet is different and everyone’s dietary requirement is different.” (Forum 14: Male, 19 years old)

#### 3.2.3. Theme 3: Transformation of Identity in Emerging Adulthood

These YA were in a self-defined transitional stage of their lives. Time spent figuring out themselves, understanding the world and their future life direction were prioritised over healthy behaviours. Some participants could perceive their attitudes and behaviours changing as they grow older, finish tertiary education, and have the knowledge, time and money necessary to adopt the healthy behaviours that they feel they are not capable of currently. Some participants have already witnessed a change in themselves and their health as they have matured and learnt from their lived experiences.

“I think the only difference in young adults is that we’re thrown into a world that we are still trying to understand, and I think it can be confusing and difficult to figure out who we are or who we want to be. I’m just hoping as I mature and experience life that I’m able to better understand how to keep myself healthy.” (Forum 4: Female, 22 years old)

This phase of their lives often involved recent independence from their parents or carers whom they had relied on for financial support and decision making. Financial independence, including budgeting and paying for rent, food and sport or gym, was an important and often new aspect of their lives. As money was often seen as a limiting factor to their choices and financial literacy was perceived as low, it was regarded as something they needed to figure out and prioritise. Some participants who were reliant on their parents, described a perceived lack of control over what food was bought and served, and therefore their eating habits. Independence, both related to finances and housing, allowed these YA to make decisions for themselves and change behaviours.

“But I believe that the full realisation of what those terms (Health and wellbeing) mean in relation to your life is only fully realised until you have complete responsibility in your life. Which is why it is sometimes hard to be healthy when you’re younger and you rely on your parents.” (Forum 4: Male, 23 years old)

These YA often described themselves in reference to others, whether those were peers or people they did not know personally, such as SM influencers. Social norms and comparisons to others shaped the development of identity, as some YA would base their attitudes, ideals and behaviours on others. SM was fuelling this comparison, as it was a platform for people to broadcast their lives and constantly compare their physical appearance, behaviours and life experiences with others, forming and transforming their identities. This, in turn, negatively affected some participants’ self-esteem and body image.

“i feel that it’s very different for young adults / teens because we have so many pressures of looking as good as someone we know that’s maybe “skinny” or more in shape than ourselves.” (Forum 4: Female, 23 years old)

SM was perceived as a constantly evolving culture of popular eating and exercise behaviours shaped by the rise of SM influencers and their idealised healthy lifestyles. These lifestyles portrayed on SM, although sometimes inconsistent with government health recommendations, were seen as ideal and something they should be following to fit in or be perceived favourably by peers. The comparison to the healthy ideal either motivated participants to change their behaviours to be more like the SM influencers or created a pressure to conform to these standards they often perceived as unachievable. Some YA reported feeling inadequate for not being able to reach these unattainable lifestyle behaviour goals.

“Too much of what is out there is coming from self-professed “gurus” and not from those with formal education. A lot of well being content is about looking “healthy” to impress people on Instagram, rather than what is actually good for you or realistic.” (Forum 4: Female, 24 years old)

#### 3.2.4. Theme 4: Promotion of Health-Enhancing Behaviours Needs to Be Different and Tailored

When participants shared their health promotion strategies, the ideals were diverse but largely addressed the identified barriers to healthy eating. The AGHE did not relate to how many of these YA view healthy eating and did not address the barriers they face, and was therefore not seen as an optimal healthy eating message to evoke behaviour change.

Many participants described campaigns with some kind of monetary “reward” for participating, or a discount for consuming more fruit and vegetables, as schemes that would encourage change.

“Incentivising them would work. The best incentive is monetary. If money wasn’t an issue i would pay people to eat fruits/vegetable regularly over 4 weeks. After that, it would hopefully be a habit and so they would do it out of their own will instead of being incentivised!” (Challenge 1: Female, 24 years old)

Others suggested the provision of free fruit and vegetables in public settings, particularly university, would encourage consumption. The collective voice reflected a struggle that these YA experience in accessing fruits and vegetables from both a cost, time and convenience perspective.

Current health messaging such as the AGHE was sometimes seen as difficult to follow, as it was too detailed, difficult to translate to actual meals, and was therefore unattainable. Some participants suggested interesting and fun ways to prepare healthy food would encourage behaviour change more than a guide such as the AGHE. It was important that these recipes were quick and inexpensive, to address both time and budget barriers. The provision of recipes also addressed the barrier some face of knowing what exactly is healthy and how to prepare these healthy foods. Furthermore, education on the benefits of fruit and vegetable consumption that mattered and was relevant to their needs and desires was suggested. The content of the messages proposed was less focussed on disease prevention and more on immediate tangible benefits such as improvement in hair, skin, physical activity capability and mental and emotional state.

“Rather than that fear factor maybe promoting the health benefits of fruit and vegetables would be more inspiring.It’s easy to say ‘vegetables and fruit are good for you’ but if a person can see exactly what the food they’re putting into their body is going to do that could be effective!” (Challenge 1: Female, 20 years old)

Healthy eating and current healthy eating messages were sometimes perceived as boring and in need of a revamp and an element of fun to increase their appeal. Humour and the use of memes were seen as appealing attention-grabbing techniques that could be utilised. The involvement of peers was commonly mentioned as another fun campaign idea which would get people involved and invested due to the desire to fit in and be with their peers. Ideas included online challenges, and sharing fruit and vegetable consumption achievements through SM and community events.

To gain exposure from a large audience, it was seen as beneficial to create a “viral” campaign related to fruit and vegetables on SM, with some mentioning the use of a catchy hashtag to help spread content. Going viral was important as it catches people’s attention, creates conversations and a new norm or culture for YA to engage in. Celebrities, SM influencers and athletes were suggested as potential spokespeople of the campaign.

“grocers should turn to sponsoring social media icons (government could do this too), particularly famous/popular instagram and youtube online celebrities who have vast followings. These should ideally be individuals who provide content that is health/food/exercise/sport/beauty centered, as the target audience would be those who have already recognised they want to lead a healthier lifestyle, who are in the process of change (or at least recognising that they need change).” (Challenge 1: Male, 22 years old)

## 4. Discussion

This qualitative analysis explored the diverse attitudes towards health and wellbeing amongst Australian YA, of whom the majority were attending tertiary education. Health was seen as a holistic interplay between physical, emotional, financial, social, cultural and spiritual wellbeing. Disease states and the long-term benefits of health-enhancing behaviours were not easy to relate to, and their suggested health communications often highlighted short-term tangible benefits of health-enhancing behaviours. Prioritisation was often placed on developing their identity, understanding their place in the world, studying, working and socialising. Participants often believed health communications should be relatable to their needs, particularly cost and convenience. Current health communications such as the AGHE did not address these needs, were seen as hard to understand or implement, and were not consistent with other healthy eating information online.

Health was discussed by young adults using holistic language, and not focused on individual behaviours such as cooking or consuming food but rather viewing other aspects of health such as mental health and wellbeing as synchronous with eating. Participants primarily used positive, gain-frame language or tone when discussing their ideas of future health campaigns. For preventive health behaviours, such as healthy eating, a positive or gain-framed message has previously been found to more effectively promote the adoption of those behaviours [40]. Similarly, previous research has shown that guilt, fear and shame-based social marketing communications aimed at changing behaviour promote inaction rather than action in some groups [41]. When engaging with young adults on social media platforms, health professionals could benefit from using this type of language, tone and communication techniques when discussing health and healthy eating.

Overall, health was viewed as something important but often not a priority at these YA’s current life stage. Previous research on YA’s and tertiary student’s healthy eating behaviours indicated there was a similar apathy towards long-term health implications [23,42] and a sense that they will start to care more about their health when they are older [23]. Similar to previous research [23,43], our findings identified barriers to eating healthily including lack of time (often due to study commitments), lack of knowledge of how to incorporate healthy food into their current lifestyle, the expense of healthy food relative to unhealthy options and convenience foods, and the influence of peers and societal norms around unhealthy food.

One aspect of health that resonated with many participants was mental health, as it was something they could personally relate to. The prevalence of mental illnesses in YA is higher than other age groups [44,45] and approximately three-quarters of mental health disorders start by the mid-twenties [46]. Mental illnesses are a concern in University students [47], and stigma surrounding these illnesses is apparent [48]. Emerging adulthood may therefore be a pivotal time to encourage wellbeing promoting certain behaviours and the maintenance of mental health. Fortunately, the YA in these analyses described a sense of openness to talk about these issues and a move away from the stigmatisation of mental illness which has previously been associated with lack of empowerment to seek treatment, discrimination and lower self-efficacy and self-esteem [49].

There was a disconnect between the holistic view of health of these YA and the usual approaches of health promotion, focussing on specific aspects of health. The YA sentiments on the AGHE echoed previous research where Government nutrition messages often fail to grasp the attention of the public and inspire behaviour change as they are viewed as incorrect, biased and inconsistent [50], or simply not relevant to them [20]. The participants also mentioned conflicting nutrition messages online and on SM as a hindrance to knowing how to eat healthily. Perceived constant exposure to nutrition information through SM [20] and the evolutionary nature of nutrition science likely contributes to misconceptions and conflicting information, as there is continually new research which may have contradictory findings. The media, including SM and influencers, draw different conclusions from nutrition research, particularly those who are not educated in nutrition science, leading to conflicting messages for the public to interpret [50].

Health communication strategies suggested by participants often highlighted the barriers they face. Some of the suggested strategies included making healthy eating cost effective, available in a location convenient to them such as university, and involving friends to make healthy eating a normative behaviour. Whereas in another study, YA were found to want nutrition information related to weight loss and serving size [20]. Previous public health initiatives have focused on promoting knowledge of fruit and vegetables, but have had a low to modest ability to sustain behaviour change [51]. Therefore, similar to those measures proposed by participants related to cost of healthy fresh food, there is now a move towards more systemic changes that influence the affordability of healthy food, such as the sugar tax and subsidising fruit and vegetables [52].

Consistent with emerging adulthood theory [17], this group of YA were in distinctly different stages of their lives, with different values, life experiences and understandings of themselves and their future. This life stage is characterised by exploration and the development of identity and values [53]. Similarly, in the current analyses, the YA were sometimes unsure of their identity and believed it would change as they aged and with life experience. The participants also described themselves as being in a transitional stage of life, with many recently becoming independent from their parents. Previous research has highlighted similar detrimental effects of this transition on health-enhancing behaviours due to the need to balance studying, work, socialising and finances [54]. Throughout young adulthood, values change and, therefore, priorities relating to food choice such as cost, convenience, taste and health also change, which then predicts changes in diet quality, such as the consumption of fruit and vegetables and sugar-sweetened beverages [55].

In the current analyses, the health-related values of these YA were often modelled through comparisons with other people and their lifestyles, particularly through SM. The YA in this these analyses described comparing themselves to others and basing their self-perception on others. This finding is consistent with social comparison theory, where people will assess their own abilities and opinions by comparing them to other people’s abilities and opinions [56]. This phenomenon is particularly prevalent in emerging adults as they are forming their identity [57], and with SM where people have a tendency to present themselves in a favourable and often unrealistic way, which followers compare themselves to [58]. In previous research, such SM use has been found to increase the tendency for people to negatively compare themselves to others [59], which can negatively affect people’s wellbeing [60], body image [61] and mood [62].

Potential limitations of the current analyses include the sample recruitment where only individuals from market research panels were invited to participate. The sample was skewed towards those who are well educated, with the majority completing tertiary education. The cultural identity of the participants was not well captured in the data collection which hinders the ability to explore different perspectives based on different cultures. The high attrition rate across the 4-week online conversation may indicate some participant fatigue in completing the conversations over an extended period. There may have been a bias towards interest in health in those completing all the activities in the online conversations compared to those who did not complete. Only those who use SM twice or more a day were eligible; however, given the rates of SM use among the Australian YA population [63], this is unlikely to have excluded many YA.

## 5. Conclusions

In conclusion, these YA need relevant normative motivation and incentive to focus on health, as the long-term consequences are not tangible or relevant to them at their particular life stage. Potential beneficial communication strategies may include reducing and dispelling inconsistent messages by increasing knowledge of healthy food and recipes that are quick and affordable, and promoting the short-term benefits of healthy behaviours such as mental wellbeing and capacity for physical activity. Tertiary education settings provide a unique opportunity to target large groups of young adults who are going through a similar life stage. The next phases of the Communicating Health Study aim to further understand communication strategies that engage YA [24]. Further phases of the study will gather additional information on the health-related values of YA, utilise market segmentation to tailor messaging, and use co-design to provide recommendations for effective health messaging strategies catered to and addressing the needs of this pivotal age group. Our application of social marketing techniques, and now the identification of the importance of communication strategies, provides direction for the next phases of the Communicating Health study [24]. In order to encourage successful behaviour change, we will incorporate the holistic nature of health that was described by participants. Future health communications or nutrition strategies should incorporate the many important aspects of life that affect the eating habits of these young adults, including money, time, knowledge of how to easily eat healthier, and social expectations around eating. To support behaviour change, there is a need to create systemic environmental changes to increase the proximity and accessibility of convenient healthy food, exercise opportunities and mental health support alongside any future health communications.

## Figures and Tables

**Figure 1 nutrients-12-00887-f001:**
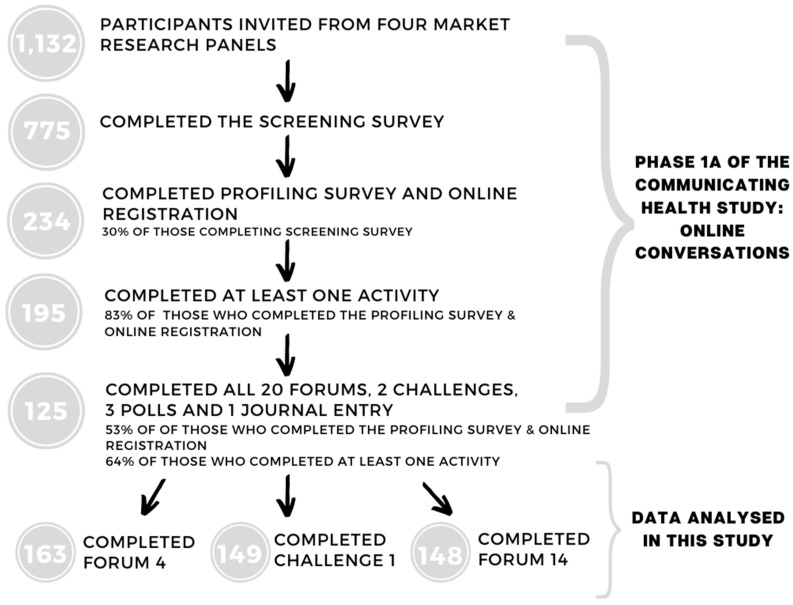
Study flow diagram.

**Table 1 nutrients-12-00887-t001:** Discussion Guide for the forums analysed.

Forum/Challenge	Discussion Guide	Logic of Inquiry
Forum 4: Healthy Lifestyle	Now this is a very broad question, so very broad answers welcome. I really want to understand how you approach the topic of ‘health and well-being’. It might be that some of you think about it a lot, it might that some of you never do and that’s fine. We just want to know how you approach it. So please share: ● What does the term ‘healthy or well-being’ mean to you? What are some of the first things that come to mind? And how do you feel when you hear this term? ● Is ‘health and well-being’ different for young adults? How? How has it changed for you over the years? ● Do you think about your health? Some of you have already shared things about your health... What other lifestyle activities are you thinking about changing or doing?For example, I know that I need to exercise more and I want to. I think about it all the time… and yet do nothing about it. ● Do you think that this will change over the next couple of years? How? Please be as detailed as you can so we can understand why you’re saying what you’re saying. Otherwise, we’ll be commenting asking for more.	To set the scene, gather participants’ perspectives on how health is framed in the young adult’s lives and explore their perceived change of perspectives over time in relation to importance, priorities and concerns during the period of emerging adulthood
Forum 14: The Healthy Eating Guide	Shown photo of the Australian Guide to Healthy eating. Asked the following questions: 1. Have you ever seen this guide before? 2. When you look at this image, what is the first thing that comes to mind? 3. Is this how you think about what you eat? 4. If you had to describe this to one of your younger siblings or cousin, how would you describe it?	To present a national healthy eating communication strategy and understand participants’ awareness and engagement with the messages; and using hypothetical teach-back scenario to gather how participants understand the healthy eating message
Challenge 1: My Campaign	Let’s get creative! Your mission is to get young adults like yourselves to take increase their daily fruit and vegetable consumption. Here’s the catch - you’re not allowed to just use advertising like TV ads/commercials or web ads - we want you to really think outside the square! Think about content videos, articles, games, competitions, celebrities and so on. Post your idea below. Once you’ve done this, please read the other ideas and like all those you find would be effective with you. GO! :)	To facilitate discussion on creating communication strategies for healthy messages that resonates with the young adults.

**Table 2 nutrients-12-00887-t002:** Participant characteristics of those completing at least one forum/challenge analysed (*n* = 163 participants).

Variable	Category	*n* Participants (% of Total)	Mean (Standard Deviation)
Age (years)			20.9 (2.2)
	18–21	91 (55.8%)	
	22–24	72 (44.2%)	
Gender identity	Female	99 (60.7%)	
	Male	63 (38.7%)	
	Non-binary/genderfluid/genderqueer	1 (0.6%)	
Body mass index (BMI) kg/m^2^			24.6 (6.0)
	Underweight (BMI < 18.5 kg/m^2^)	17 (10.4%)	
	Healthy weight (BMI 18.5–24.9 kg/m^2^)	88 (54.0%)	
	Overweight (BMI 25.0–29.9 kg/m^2^)	35 (21.5%)	
	Obese (BMI ≥ 30.0 kg/m^2^)	23 (14.1%)	
Currently studying	No	53 (32.5%)	
	Yes	110 (67.5%)	
Level of current study	High school, year 12	8 (4.9%)	
	TAFE, college or diploma	12 (7.4%)	
	University (undergraduate course)	80 (49.1%)	
	University (postgraduate course)	10 (6.1%)	
Highest level of completed education	High school, year 10 or lower	1 (0.6%)	
	High school, year 11	2 (1.2%)	
	High school, year 12	12 (7.4%)	
	TAFE, college or diploma	21 (12.9%)	
	University (undergraduate degree)	15 (9.2%)	
	University (postgraduate degree)	2 (1.2%)	
Location *	Metro	130 (79.8%)	
	Regional/rural	33 (20.2%)	
Living arrangements †	Alone	19 (11.7%)	
	Living with own child(ren)	17 (10.4%)	
	Other family	16 (9.8%)	
	Friend(s)/housemate(s)	26 (16.0%)	
	My partner	35 (21.5%)	
	Living with parents	79 (48.5%)	
Dispensable weekly income ‡	Less than $AU40	64 (39.3%)	
	$AU40–$79	48 (29.4%)	
	$AU80–$119	29 (17.8%)	
	$AU120–$199	11 6.7%)	
	$AU200–$299	8 (4.9%)	
	$AU300 or over	2 (1.2%)	
	I don’t wish to say	1 (0.6%)	
Language spoken at home	English	121 (74.2%)	
	Language other than English	42 (25.8%)	

* Location question: “Please confirm where you live: 1. Sydney metro area; 2. Other New South Wales (regional/rural); 3. Melbourne metro area; 4. Other Victoria (regional/rural); 5. Brisbane metro area; 6. Other Queensland (regional/rural); 7. Adelaide metro area; 8. Other South Australia (regional/rural); 9. Perth metro area; 10. Other Western Australia (regional/rural); 11. Hobart metro area; 12. Other Tasmania (regional/rural); 13. Australian Capital Territory (Metro); 14. Northern Territory (regional/rural)”. † Participants could choose more than one response. ‡ Dispensable weekly income question: “During a normal week, how much money do you have to spend on yourself for recreational purposes?”.

**Table 3 nutrients-12-00887-t003:** Themes and brief descriptions from analysis of complete data set.

Theme	Description
Theme 1: Perceptions of health and wellbeing through the lens of emerging adults	Overall participants described health as a balance between their physical, emotional, financial, social, cultural and spiritual states. Mental health was highlighted as an important aspect of health, often due to being personally affected by mental health issues. Many stated they were at optimal health status however were at different stages of engagement with health-enhancing behaviours. Short-term tangible often gender-specific benefits—such as appearance and physical activity capacity improvements from health-enhancing behaviours—were more relatable than long-term.
Theme 2: Competing demands and priorities limiting health-enhancing behaviours	A prominent barrier to performing health-enhancing behaviours was the financial costs of eating healthily, participating in organised exercise and mental health services. Participants also described their lack of time to perform these behaviours due to the prioritisation of other aspects of their lives such as studying, working and socialising. Some participants also described a lack of knowledge and the prevalence of misconceptions around what is actually healthy.
Theme 3: Transformation of identity in emerging adulthood	Participants were in a self-defined transitional life stage where they were figuring out their identity and creating values. There was prioritisation of identity development over performing health-enhancing behaviours. Many described recent independence, which resulted in challenges related to financial literacy and independent decision making. Participants often described comparing themselves to others particularly via idealised lifestyles portrayed on social media. This comparison either motivated behaviour change or created a pressure to conform.
Theme 4: Promotion of health-enhancing behaviours needs to be different and tailored	Current healthy eating messages such as the AGHE do not address the barriers they face to healthy eating. The most common strategies described to encourage fruit and vegetables involved incentives and peers. Strategies to address barriers to healthy eating were addressed in the form of free and/or convenient access to healthy food in places they frequent such as university. To grab and maintain attention it was suggested that strategies should provide relevant messages such as benefits of health-enhancing behaviours beyond disease prevention and information to dispel inconsistent messages.

AGHE: Australian Guide to Healthy Eating.

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
