# Peer review of "Language of Health of Young Australian Adults: A Qualitative Exploration of Perceptions of Health, Wellbeing and Health Promotion via Online Conversations"

_nutrients, 2020, doi:10.3390/nu12040887_

Round 1

Reviewer 1 Report

This is an interesting study focusing on young adults’ perceptions related to health, healthy behaviours and health promotion efforts using a qualitative approach. I have made several suggestions that the Authors may want to consider in revising their manuscript.

Introduction

Lines 54-55: Emerging Adulthood theory should be expanded on. At present, this sentence does not add many value to the information presented in the introduction. What kind of health-enhancing behaviours Arnett et al. have proposed?

The hypotheses should be formulated in the introduction section.

The discussion needs to go more in depth in order to develop a better understanding of language of health of young adults. A more detailed outlook on the continuation of the study considering the results should be presented. In addition, Authors should express an opinion about the practical implications of the findings.

Appendix: Figure A1 is not very readable (e.g. the variable “change”, “healthier”; the variable between “media” and “people” is illegible). Figure A1 should be improved.

Author Response

Point 1: This is an interesting study focusing on young adults’ perceptions related to health, healthy behaviours and health promotion efforts using a qualitative approach. I have made several suggestions that the Authors may want to consider in revising their manuscript.

Response 1: We wish to thank you for reviewing our manuscript and for providing us with useful advice to strengthen the reporting of our research.

Point 2: Lines 54-55: Emerging Adulthood theory should be expanded on. At present, this sentence does not add many value to the information presented in the introduction. What kind of health-enhancing behaviours Arnett et al. have proposed?

Response 2: We agree this should be expanded so have added information as follows “This has been described by Arnett et al. as Emerging Adulthood theory [17] and is characterised by the development of identity and values related to different areas of life including love, work, worldviews and risky behaviours. We postulate that this could include the value placed on health-enhancing behaviours and the formation of habits related to these behaviours such as substance use, eating and physical activity, therefore, targeting this potentially malleable age group could be an impactful health promotion effort [1].” (Lines 57-62 **NB line number refer to TRACK CHANGES version of manuscript)

Point 3: The hypotheses should be formulated in the introduction section.

Response 3: This research was formative exploratory research used to inform future more hypothesis-driven research. We have added in an explanation of this as follows: “The purpose of this research was to provide formative information to inform future more hypothesis-driven research and was therefore exploratory in nature.” (Lines 82-84)

Point 4: The discussion needs to go more in depth in order to develop a better understanding of language of health of young adults. 

Response 4: We have attempted to address your comment with the additional information following: “Health was discussed by young adults using holistic language, and not focused on individual behaviours such as cooking or consuming food but rather viewing other aspects of health such as mental health and wellbeing as synchronous with eating. Participants primarily used positive, gain-frame language or tone when discussing their ideas of future health campaigns. For preventive health behaviours, such as healthy eating, a positive or gain-framed message has previously been found to more effectively promote the adoption of those behaviours [40]. Similarly, previous research has shown that guilt, fear and shame based social marketing communications aimed at changing behaviour, promote inaction rather than action in some groups [41]. When engaging with young adults on social media platforms, health professionals could benefit from using this type of language, tone and communication techniques when discussing health and healthy eating.” (Lines  413-422)

Point 5: A more detailed outlook on the continuation of the study considering the results should be presented. In addition, Authors should express an opinion about the practical implications of the findings.

Response 5: We hope we have addressed your comments with the following: “Our application of social marketing techniques, and now the identification of the importance of communication strategies, provides direction for the next phases of the Communicating Health study [24]. In order to encourage successful behavior change, we will incorporate the holistic nature of health that was described by participants. Future health communications or nutrition strategies should incorporate the many important aspects of life that affect the eating habits of these young adults, this includes money, time, knowledge of how to easily eat healthier and social expectations around eating. To support behaviour change there is a need to create systemic environmental changes to increase proximity and accessibility of convenient healthy food, exercise opportunities and mental health support alongside any future health communications.” (lines 503-512)

Point 6: Appendix: Figure A1 is not very readable (e.g. the variable “change”, “healthier”; the variable between “media” and “people” is illegible). Figure A1 should be improved.

Response 6: We agree that these figures should be more clear however these are the outputs that Leximancer provides. We have added in text to clarify the illegible words within this figure. (Lines 537-539)

Reviewer 2 Report

This paper explores young adults’ perceptions of health, wellbeing and health promotion using online conversations as data. The findings are interesting and important to inform future health promotion campaigns aimed at this target group. I have rated this paper as low interest to readers not because it is not interesting for me and many others to read but because I think it would be more suited to a health promotion or public health journal rather than a nutrition specific focused journal, as the aim was to explore health and wellbeing more widely and ways of communicating health messages more effectively in young people than just healthy eating perceptions and strategies specifically (albeit that was one particular area explored). If the Editors are happy to include this article in their journal then it requires very little editing.

Specific feedback follows:

Title: include this was a sample of Australian young people.

Abstract

  • The = sign is missing between the n and the sample size number.
  • The 2nd sentence should include the word ‘Australian’ to make it clear that this is a cohort from this country.
  • Add details of where and how the sample was recruited (eg were they recruited online or was it a sample of University students recruited through a single or multiple Universities).
  • The theories used to underpin the analysis should be mentioned in the abstract

Introduction

The final section rationale discusses specifically University student health but the methods section then talks about a sampling method to reach a representative sample across the Australian population. I suggest in line 72 adding that the sample were drawn from market research and primarily in tertiary education – this then makes it clear why literature on University students has been included in the previous sentence.

Methods

  • Under the recruitment heading, the = is missing when describing the recruitment target size (should be N=200).
  • How were people recruited into the panel? They were part of the field house’s data base but how were they recruited into that database. Is it important to comment on this in regards to the representativeness of the sample. Did every young person in Australia have the opportunity to sign up and become a panel member or was a targeted approach to recruiting young people used and if so how was this done (through what recruitment strategies – was it through social media or other avenues)?

Discussion

  • This section discusses implications of the findings well and in the context of current relevant literature.
  • Limitations section: The sample is skewed towards those that are well educated (nearly half of the sample were undertaking an undergraduate degree). It is mentioned that participants were recruited through market research panels but it is worth mentioning again here that most were University students. Also it should also be highlighted that culture/ethnicity was not captured in data collection. Differences in perceptions based on culture and to comment on the cultural diversity of the sample would have been really useful (eg was the sample generally representative in terms of their characteristics of the large cultural diversity in Australia, acknowledging that qualitative research is not to be generalisable).

Author Response

Point 1: This paper explores young adults’ perceptions of health, wellbeing and health promotion using online conversations as data. The findings are interesting and important to inform future health promotion campaigns aimed at this target group. I have rated this paper as low interest to readers not because it is not interesting for me and many others to read but because I think it would be more suited to a health promotion or public health journal rather than a nutrition specific focused journal, as the aim was to explore health and wellbeing more widely and ways of communicating health messages more effectively in young people than just healthy eating perceptions and strategies specifically (albeit that was one particular area explored). If the Editors are happy to include this article in their journal then it requires very little editing.

Response 1: We wish to thank you for reviewing our manuscript and for providing us with useful feedback to improve our reporting. Whilst Nutrients is perhaps not an obvious choice for this type of work, we believe that the context of the special issue will attract readership to an area that is very important for all nutrition professionals to consider, particularly in the current climate of the proliferation of misinformation. Given this, We believe the interdisciplinary nature of our work is important for those within the wider research area of nutrition. Furthermore, highlighting the multifactorial viewpoint of young adults relating to healthy eating and health and the interplay between eating and other aspects of health and life is important in the context of intervention development.

Point 2: Title: include this was a sample of Australian young people.

Response 2: We have now included Australian in our title as follows: “Language of health of young Australian adults: Qualitative exploration of perceptions of health, wellbeing and health promotion via online conversations”

Point 3: Abstract. The = sign is missing between the n and the sample size number.

Response 3: Thank you for pointing this out, have now added in the = sign (Line 21).

Point 4: The 2nd sentence should include the word ‘Australian’ to make it clear that this is a cohort from this country.

Response 4: We have now added in Australian as follows “This research explored Australian YA values and perceptions related to health, healthy behaviours and health promotion efforts.” (Lines 19-20)

Point 5: Add details of where and how the sample was recruited (eg were they recruited online or was it a sample of University students recruited through a single or multiple Universities).

Response 5: We have now added in more detail “This qualitative analysis involved n=166, Australian 18-24 year olds recruited through a market research field house.” (Lines 20-21)

Point 6: The theories used to underpin the analysis should be mentioned in the abstract

Response 6: We have now added in those theories “LeximancerTM and manual inductive thematic coding were utilised for analysis via the lens of emerging adulthood and health communication theories.” (lines 23-25) 

Point 7: Introduction. The final section rationale discusses specifically University student health but the methods section then talks about a sampling method to reach a representative sample across the Australian population. I suggest in line 72 adding that the sample were drawn from market research and primarily in tertiary education – this then makes it clear why literature on University students has been included in the previous sentence.

Response 7: Thank you, we have now added in the following to explain this. “Therefore, the primary objective of this research was to explore what health, wellbeing and healthy eating mean to a sample of Australian YA aged 18 to 24 years old who were primarily completing tertiary education recruited through a market research field house, and what aspects of health are important to them at this stage of their lives.” (Lines 76-79)

Point 8: Methods. Under the recruitment heading, the = is missing when describing the recruitment target size (should be N=200).

Response 8: Thank you we have added that. (Line 108)

Point 9: How were people recruited into the panel? They were part of the field house’s data base but how were they recruited into that database. Is it important to comment on this in regards to the representativeness of the sample. Did every young person in Australia have the opportunity to sign up and become a panel member or was a targeted approach to recruiting young people used and if so how was this done (through what recruitment strategies – was it through social media or other avenues)?

Response 9: We agree this should be explained more explicitly so have altered. “Panel members were from a mix of panels and were already part of the field house’s database and voluntarily provided their details in the expectation they would be invited to partake in research. The field house recruited panel members for the purpose of research, not specifically for this study. Research panels are recruited through online advertising and social media targeted at a range of individuals not specifically young adults and anyone is eligible to sign up to a panel. The online conversations study was advertised to those within these panels through the online panel research websites.” (Lines 115-121). 

Point 10: Discussion. This section discusses implications of the findings well and in the context of current relevant literature.

Response 10: Thank you for your feedback.

Point 11: Limitations section: The sample is skewed towards those that are well educated (nearly half of the sample were undertaking an undergraduate degree). It is mentioned that participants were recruited through market research panels but it is worth mentioning again here that most were University students. 

Response 11: Thank you we agree this should be highlighted. “The sample was skewed towards those who are well educated with the majority completing tertiary education.” (Lines 482-483)

Point 12: Also it should also be highlighted that culture/ethnicity was not captured in data collection. Differences in perceptions based on culture and to comment on the cultural diversity of the sample would have been really useful (eg was the sample generally representative in terms of their characteristics of the large cultural diversity in Australia, acknowledging that qualitative research is not to be generalisable).

Response 12: Thank you we agree that we should highlight the limitation of not exploring cultural diversity in our sample. “The cultural identity of the participants was not well captured in data collection which hinders the ability to explore different perspectives based on different cultures.” (Lines 483-485)